## PERSPECTIVE

# Unravelling the complexities of neuromuscular function in females throughout the adult lifespan

**J. Greig Inglis** 🆔 **and Hélio V. Cabral**

*Department of Clinical and Experimental Sciences, Università degli Studi di Brescia, Brescia, Italy*

Email: james.inglis@unibs.it

Handling Editors: Richard Carson & Christoph Centner

The peer review history is available in the Supporting Information section of this article (https://doi.org/10.1113/JP289013#support-information-section).

Neuromuscular function encompasses the integration and translation of synaptic potentials to the alpha-motoneurons, which activate muscles. Muscles will subsequently shorten and transmit the tension down the tendons' line of action, resulting in joint mechanical movement. Unfortunately, neuromuscular performance deteriorates via multiple factors throughout the adult lifespan. As has been reported previously, age can significantly impact the neuromuscular system's ability to generate maximal force, more so in ageing females approaching and surpassing the menopause (Piasecki et al., 2024).

In the current issue of *The Journal of Physiology*, O'Bryan et al. (2026) have taken steps to close this important knowledge gap in terms of the vulnerability and changes in neuromuscular function with ageing, specifically in adult females throughout the adult lifespan. O'Bryan et al. (2026) report a quasi-linear decrease in maximal compound muscle action potential (M-wave amplitude) in the rectus femoris with age without similar changes in the vastii, which suggests a vulnerability of the bi-articular rectus femoris to age-related impairments in neuromuscular function across the female lifespan. Generally, the rectus femoris has more type II fibres compared to the vastii, which would primarily atrophy with age. Furthermore, the rectus femoris $Na^+$ channels have been reported to become less effective in older females, which was attributed to down-regulation of $Na^+/K^+$ ATPase, contributing to denervation. Additionally, the rectus

femoris appears to atrophy before the vastii, therefore, a reduced M-wave amplitude of the rectus femoris is probably caused by both a reduction in the lean muscle cross-sectional area and denervation with age. The significance of these findings is that the rectus femoris plays a significant role in torque generation and transfer across the hip and knee joints during locomotion functional tasks, which is specifically important in older adult females who have a greater risk of falls (O'Bryan et al., 2026).

It is admirable that O'Bryan et al. (2026) have taken on the challenge to unravel the complexities of the effect the menopause stages have on neuromuscular function. The dramatic hormonal changes associated with the stages of the menopause can result in a greater loss in force-generating capacity of the muscle in humans. Despite this, pre-, peri- and post-menopausal human females are vastly understudied in terms of neuromuscular function, which is likely a direct result of these complexities and variations in sex hormone concentrations within each given cohort. Therefore, the study by O'Bryan et al. (2026) fills a large knowledge gap in this area of study and will hopefully serve as a catalyst for further research in this area.

O'Bryan et al. (2026) have significantly contributed to filling an important knowledge gap in the literature regarding the adult female lifespan. The large data set across many participants ($N = 88$), representing the stages of the menopause (peri, pre and post), and many measures provide clarity in the neuromuscular function of this understudied group. The study included several associated measures across all phases of the adult female lifespan, including sex hormone concentrations, lean skeletal muscle cross-sectional area, intramuscular adipose tissue, specific force, evoked torque and electromyographic outcomes. These measures provide scoping insights into the complexities involved in investigating this vastly understudied population. As identified by O'Bryan et al. (2026), and not their sole measure of neuromuscular function, the use of global electromyography amplitude as a measure of neuromuscular function has several limitations and leaves room for further investigations to explore this area using the evolving high-density surface electromyography (HDsEMG) technology

and associated decomposition algorithms. A review by Piasecki et al. (2024) and a research paper by Guo et al. (2025) explored female ageing and neuromuscular function. In these studies, there were distinguishable differences in motor unit characteristics of older women and men, as assessed by intramuscular electromyography and HDsEMG, that suggest changes in neuromuscular control during the adult female lifespan. Specifically, greater motor unit discharge rates and reduced force steadiness were observed in older females, which may be related to the modulation of the motor unit discharge rate and changes in joint laxity (Inglis & Gabriel, 2021). As seen in the study by O'Bryan et al. (2026), during the menopause, there is a reduction in muscle quality seen in the increase in intramuscular adipose tissue, which was supported by the accelerated reductions in voluntary and evoked force responses during the menopausal transition period. O'Bryan et al. (2026) also suggested that changes in motor unit discharge rates may be linked to the changes in muscle quality and that future research should address this association. Similarly, this was also reported as a neural strategy by Guo et al. (2025), who suggested that, as a result of the greater myosteatosis (reduction in muscle quality), ageing females may have to increase their discharge rate to achieve the same force outputs as their male counterparts.

In conclusion, the study conducted by O'Bryan et al. (2026) provides valuable insights into muscle physiology and neuromuscular adaptations during the female adult lifespan, demonstrating that the onset of neuromuscular degeneration from the onset of the menopause initially targets intrinsic peripheral muscle function prior to functional declines. Their study provides a significant contribution and fills several knowledge gaps in the literature in this vastly understudied area. Future research will need to address this noted association between hormonal changes during the menopause and the associated changes in muscle loss and muscle quality with the addition of how these changes may influence motor unit behaviour. Additionally, these changes in hormone status and motor unit characteristics will need to be compared with changes in muscle quality and the potential effects at the cellular level focused on the regulation

of the Na$^+$/K$^+$ pump. Furthermore, the inclusion of motor unit coherence analysis in this population with the aid of non-invasive motor unit tracking HDsEMG could highlight some of the changes throughout the menopause in the common synaptic inputs to spinal motoneurons that are a determinant of force output steadiness. Additionally, the assessment of the effects of changes in muscle quality in this population may be utilized for specific rehabilitation modalities tailored to the stages of the menopause as needed to reduce falls with age. These findings would become the cornerstone for promoting ageing women's health as they would apply to the prescription of sex-specific interventions, which may be required earlier in the adult lifespan compared to their male counterparts.

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

## Additional information

### Competing interests

No competing interests declared.

### Author contributions

J.G.I and H.C conceived or designed the work and drafted the work or revised it critically for important intellectual content. Both authors approved the version final version of the manuscript submitted for publication and agree to be accountable for all aspects of the work.

### Funding

EC | Horizon Europe | Excellent Science | HORIZON EUROPE Marie Skłodowska-Curie Actions (MSCA): J. Greig Inglis, 101151712 J Greig Inglis was supported by the Marie Skłodowska-Curie Actions Grant 'MUDecomp' agreement no. 101151712.

### Acknowledgements

J. Greig Inglis was supported by the Marie Skłodowska-Curie Actions Grant 'MUDecomp' agreement no. 101151712.

### Keywords

discharge rate, force, menopause, motor unit, muscle activation, muscle quality, neuromuscular function, rate coding

### Supporting information

Additional supporting information can be found online in the Supporting Information section at the end of the HTML view of the article. Supporting information files available:

**Peer Review History**

