## [Peer Review History · The Journal of Physiology]

Unravelling the complexities of neuromuscular function in females throughout the adult lifespan.

J. Greig Inglis and Helio V Cabral
DOI: 10.1113/JP289013

Corresponding author(s): J. Greig Inglis (james.inglis@unibs.it)

The following individual(s) involved in review of this submission have agreed to reveal their identity: Steven Jeffery O'Bryan (Referee #1)

Review Timeline:

Submission Date:	29-Apr-2025
Editorial Decision:	13-May-2025
Revision Received:	22-May-2025
Editorial Decision:	28-May-2025
Revision Received:	28-May-2025
Accepted:	06-Jun-2025

Senior Editor: Richard Carson

Reviewing Editor: Christoph Centner

Transaction Report:

Dear Dr Inglis,

Re: JP-P-2025-289013 "Unraveling the complexities of neuromuscular function in females throughout the adult hormonal lifespan" by J. Greig Inglis and Helio V Cabral

Thank you for submitting your manuscript to The Journal of Physiology. It has been assessed by a Reviewing Editor and by 1 expert referee and we are pleased to tell you that it is acceptable for publication following satisfactory major revision.

Please advise your co-author of this decision as soon as possible.

The review comments are copied at the end of this email.

REVISION CHECKLIST:

We look forward to receiving your revised submission.

Yours sincerely,

Richard Carson
Senior Editor
The Journal of Physiology

EDITOR COMMENTS

Reviewing Editor:

We appreciate your submission and the effort that you have put into drafting the perspective article. Based on the expert reviewer's comments, we recommend that you provide additional clarity and expand upon new perspectives that go beyond paraphrasing the original article. Specifically, please address the need for High-Density Electromyography (HDEMG) and muscle quality in a way that enhances the discussion, and ensure that the interpretations of previous research are thoroughly clarified.

REFEREE COMMENTS

Referee #1:

We thank the authors for taking the time to read and provide positive commentary on our recently accepted paper in The Journal of Physiology. There are some aspects and comments which we believe require some clarification.

Comment #1

Title (and throughout): Use of the term 'hormonal lifespan' may suggest it does not include postmenopausal females, but our study included females up to 80 years of age. The authors may like to consider if it is appropriate or necessary using the word 'hormonal' here and the term 'hormonal lifespan' throughout.

Comment #2

Line 36 - 41: It is written that hormonal changes associated with menopause can reduce the force generating capacity of the muscle, and that this is a "well known fact". As we described in the second paragraph of our introduction, the cause-and-effect relationships between steroidal sex hormones and neuromuscular function in animal studies is clear, but no human studies have concomitantly evaluated muscle function and sex hormone concentrations across different phases of the lifespan, which then provided the rationale for our study. The authors may like to consider either removing this sentence or re-wording and using it later in the perspective article as an outcome from our research.

Comment #3

Line 51: Please re-word "were shown". We did not show less effective Na⁺ channels in rectus femoris of older females, rather, we referenced previous research to discuss the measured reduction in rectus femoris M-wave amplitude.

Comment #4

Line 67 - 70: We feel these lines read as though the amplitude of the bipolar surface EMG signal was the only measure we implemented to assess neuromuscular function, which is not correct. The language in this paragraph also sounds negative (e.g. "although thorough", "nevertheless"). Perhaps the authors may like to consider if they could emphasize that we also implemented high and low frequency stimulation to accompany these measures and to provide additional details pertaining to neuromuscular function, and that we took all possible steps to increase the validity of the amplitude of the voluntary surface EMG signal by normalizing it to the maximal M-wave. Perhaps the authors could also mention that in our article we agree "more research adopting high-density surface electromyography is required in females" (lines 697-698) - so the language could be changed to illustrate that we have considered and acknowledged this point. We agree that this approach is needed and will add further advancements in this field.

Comment #5

Lines 78-79: Here it is stated that the study by Guo (2025) reported reduced motor unit numbers in older females compared to males. Can the authors please clarify this statement as (we may be missing it) the paper by Guo (2025) did not seem to quantify motor unit numbers. Do the authors mean less motor units were sampled, or that there are less motor units overall? If it is the former, the relevance in the context of motor unit ageing should be clarified, and the study by Guo (2025) seemed to sample more motor units in females compared to males, not less. If it is the latter, please include a different reference or clarify where Guo (2025) reported reduced motor unit numbers in older females. Although older articles (and with some limitations highlighted by Duchateau and Enoka (2022)), Doherty (1993) and Miller (1992) showed no significant sex differences in motor unit number estimates between both younger and older males and females, despite a decrease with age.

Miller, A. E. J., MacDougall, J. D., Tarnopolsky, M. A., & Sale, D. G. (1993). Gender differences in strength and muscle fiber characteristics. *European journal of applied physiology and occupational physiology*, 66, 254-262.

Doherty, T. J., Vandervoort, A. A., Taylor, A. W., & Brown, W. F. (1993). Effects of motor unit losses on strength in older men and women. *Journal of Applied Physiology*, 74(2), 868-874.

Duchateau, J., & Enoka, R. M. (2022). Distribution of motor unit properties across human muscles. *Journal of Applied Physiology*, 132(1), 1-13.

Comment #6

Lines 79-87: Perhaps the authors could acknowledge that the relationship between muscle quality and MUFR was originally detailed in our article (lines 524-526) to demonstrate we had considered this point and the article by Guo (2025) (although muscle quality was not measured in that study). Also, please clarify that we tended to see an increase in intramuscular fat infiltration during menopause; specific force during MVC did not change, which may also be an indicator of muscle quality.

Comment #7

Lines 87-91: Like comment #5 - please clarify or reference research which shows lower motor unit numbers in older females, and also as suggested here, younger females, compared to males. The references by Guo (2022) and Yacyshyn (2025) do not seem to quantify motor unit numbers. Like above, if it is the number of decomposed units the authors are referring to please specify how that supports the statements made here.

Comment #8

Paragraph 3 lines 60-91: The authors reference several articles which use HDEMG or intramuscular EMG to investigate neuromuscular function. Despite our limitations of not implementing HDEMG or iEMG techniques, perhaps the authors may like to consider highlighting some of the strengths and the originality of our study and how it supports these previous findings. For example, we included several concomitant measures across all phases of the female lifespan which these studies did not, including sex hormone concentrations, lean skeletal muscle cross-sectional area and intramuscular fat (via pQCT), specific force, and evoked torque and electromyography outcomes. As currently written, the strengths of our study seem lost here and focus is on the omission of more direct motor unit investigation.

Comment #9

Lines 92-95: Please amend this sentence as we did not state that peripheral changes overcome reduced muscle quality. MVC, e1RM, RTD, PT10 and PT100 all decreased during the fourth decade, but changes in specific MVC and intramuscular adipose tissue were not significant (although IMAT tended to increase from 30-70 years). Specific e1RM did decrease with age but it was linear and not obvious to the menopausal transition.

Comment #10

Line 62 and 95-96: We feel that "first-step" and "starting point" somewhat undersells the depth of our research. We agree that direct investigation into motor unit behavior using HDEMG is necessary (again, which we already stated in our article), however, most other suggested aspects listed in the conclusion by the authors were incorporated into our study and were in

addition to previous research including lean skeletal muscle mass, muscle quality (via intramuscular fat infiltration and specific force), muscle function (with voluntary and evoked torque and electromyography measures using twitch interpolation plus one-repetition maximum), sex hormone concentrations, and the menopausal transition. Our study is the first to adopt this extensive and elaborate approach. Perhaps the authors may like to consider amending the language so it is more positive and highlighting more strengths of our study.

Comment #11

Line 117-119: The title of the final revised and published article is 'The Contribution of Age and Sex Hormones to Female Neuromuscular Function Across the Adult Lifespan'.

END OF COMMENTS

We thank both the editor and the reviewer for the comments below. We have clarified or modified below all aspects of our perspective on this important work in a way that improved the message for the reader while maintaining the limits (900 words, 5 references). Please find below our specific responses.

EDITOR COMMENTS

Reviewing Editor:

We appreciate your submission and the effort that you have put into drafting the perspective article. Based on the expert reviewer's comments, we recommend that you provide additional clarity and expand upon new perspectives that go beyond paraphrasing the original article. Specifically, please address the need for High-Density Electromyography (HDEMg) and muscle quality in a way that enhances the discussion and ensure that the interpretations of previous research are thoroughly clarified.

Thank you for this comment we have addressed these issues which you can find specifically below as well as adding some reasoning for the necessity for future papers in this area to address changes in motor unit behaviour and also changes in muscle quality throughout the menopause. Additionally, we have added specific instances where the use of HDsEMG and muscle quality need to be utilized to address the shortcomings in this area of study.

Page, 5, lines 107-114: "...regulation of the Na⁺/K⁺ pump. **Furthermore, the inclusion of motor unit coherence analysis in this population with the aid of non-invasive motor unit tracking (HDsEMG) could highlight some of the changes throughout the menopause in the common synaptic inputs to spinal motoneurons which are determinant to the force output steadiness. Additionally, the assessment of the effects of changes in muscle quality in this population may be utilized for specific rehabilitation modalities tailored to the stages of the menopause as needed to reduce falls with age.** These findings..."

REFEREE COMMENTS

Referee #1:

We thank the authors for taking the time to read and provide positive commentary on our recently accepted paper in The Journal of Physiology. There are some aspects and comments which we believe require some clarification.

Comment #1

Title (and throughout): Use of the term 'hormonal lifespan' may suggest it does not include postmenopausal females, but our study included females up to 80 years of age. The authors may like to consider if it is appropriate or necessary using the word 'hormonal' here and the term 'hormonal lifespan' throughout.

Thank you for this comment we have addressed this in both the title and throughout the paper. Please see below:

Page 1, line 5-6: "Unraveling the complexities of neuromuscular function in females throughout the adult **hormonal** lifespan."

Page 2, line 39: "...adult **hormonal** lifespan. O'Bryan..."

Comment #2

Line 36 - 41: It is written that hormonal changes associated with menopause can reduce the force generating capacity of the muscle, and that this is a "well known fact". As we described in the second paragraph of our introduction, the cause-and-effect relationships between steroidal sex hormones and neuromuscular function in animal studies is clear, but no human studies have concomitantly evaluated muscle function and sex hormone concentrations across different phases of the lifespan, which then provided the rationale for our study. The authors may like to consider either removing this sentence or re-wording and using it later in the perspective article as an outcome from our research.

We thank the reviewer for the comment. We have reworded and placed these sentences below to support the need for this work, and further work in this vastly understudied area in humans.

Page 3, lines 56-63: "The dramatic hormonal changes associated with the stages of the menopause can result in a greater loss in force-generating capacity of the muscle, which is even more rapid after the 8th decade **in humans**. Despite this well-known fact, pre-, peri-, and post-menopausal ~~women~~ **human females** are vastly understudied in terms of neuromuscular function, which is likely a direct result of these complexities and variations in sex hormone concentrations within each given cohort. **Therefore, the article by O'Bryan and associates (2025) fills a large knowledge gap in this area of study and will hopefully serve as a catalyst for further research in this area.**"

Comment #3

Line 51: Please re-word "were shown". We did not show less effective Na⁺ channels in rectus femoris of older females, rather, we referenced previous research to discuss the measured reduction in rectus femoris M-wave amplitude.

Changed as requested.

Page 2, line 45: "...the rectus femoris Na⁺ channels **have been reported** in older females..."

Comment #4

Line 67 - 70: We feel these lines read as though the amplitude of the bipolar surface EMG signal was the only measure we implemented to assess neuromuscular function, which is not correct. The language in this paragraph also sounds negative (e.g. "although thorough", "nevertheless"). Perhaps the authors may like to consider if they could emphasize that we also implemented high and low frequency stimulation to accompany these measures and to provide additional details pertaining to neuromuscular function, and that we took all possible steps to increase the validity of the amplitude of the voluntary surface EMG signal by normalizing it to the maximal M-wave. Perhaps the authors could also mention that in our article we agree "more research adopting high-density surface electromyography is required in females" (lines 697-698) - so the language could be changed to illustrate that we

have considered and acknowledged this point. We agree that this approach is needed and will add further advancements in this field.

Thank you for your comment and we agree and have adjusted the lines to highlight the importance of the work and clarified the scope and discussion within the paper.

Pages 3 – 4, lines 61-75: "...concentrations within each given cohort. **Therefore, the article by O'Bryan and associates (2025) fills a large knowledge gap in this area of study and will hopefully serve as a catalyst for further research in this area.**

The article by O'Bryan and associates (2025) is a significant contribution to fill an important knowledge gap in the literature regarding the adult female lifespan. The large data set across many participants (N=88), representing the stages of the menopause (peri, pre and post), and many measures provide clarity in the neuromuscular function of this understudied group. The study included several associated measures across all phases of the adult female lifespan including sex hormone concentrations, lean skeletal muscle cross-sectional area, intramuscular adipose tissue, specific force, evoked torque and electromyographic outcomes. These measures provide scoping insights into the complexities involved in investigating this vastly understudied population. As identified by the authors, and not their sole measure of neuromuscular function, the use of global electromyography amplitude as a measure..."

Comment #5

Lines 78-79: Here it is stated that the study by Guo (2025) reported reduced motor unit numbers in older females compared to males. Can the authors please clarify this statement as (we may be missing it) the paper by Guo (2025) did not seem to quantify motor unit numbers. Do the authors mean less motor units were sampled, or that there are less motor units overall? If it is the former, the relevance in the context of motor unit ageing should be clarified, and the study by Guo (2025) seemed to sample more motor units in females compared to males, not less. If it is the latter, please include a different reference or clarify where Guo (2025) reported reduced motor unit numbers in older females. Although older articles (and with some limitations highlighted by Duchateau and Enoka (2022)), Doherty (1993) and Miller (1992) showed no significant sex differences in motor unit number estimates between both younger and older males and females, despite a decrease with age.

Miller, A. E. J., MacDougall, J. D., Tarnopolsky, M. A., & Sale, D. G. (1993). Gender differences in strength and muscle fiber characteristics. *European journal of applied physiology and occupational physiology*, 66, 254-262.

Doherty, T. J., Vandervoort, A. A., Taylor, A. W., & Brown, W. F. (1993). Effects of motor unit losses on strength in older men and women. *Journal of Applied Physiology*, 74(2), 868-874.

Duchateau, J., & Enoka, R. M. (2022). Distribution of motor unit properties across human muscles. *Journal of Applied Physiology*, 132(1), 1-13.

Thank you again for this comment, we have removed the reference to motor unit number as indeed it is not relevant and can be ambiguous between motor unit population and the number of identifiable motor units, and to save space due to word count restrictions.

Comment #6

Lines 79-87: Perhaps the authors could acknowledge that the relationship between muscle quality and MUFR was originally detailed in our article (lines 524-526) to demonstrate we had considered this point and the article by Guo (2025) (although muscle quality was not measured in that study). Also, please clarify that we tended to see an increase in intramuscular fat infiltration during menopause; specific force during MVC did not change, which may also be an indicator of muscle quality.

Thank you for your comment and we have adjusted these lines to highlight your suggestion of MUDR and muscle quality relationship and your inclusion of the increase in intramuscular fat with menopause.

Page 4, lines 88-92: "...muscle quality **seen in the increase in intramuscular adipose tissue; however, they did not find any significant changes in force output. O'Bryan and associates (2025) also suggested changes in motor unit discharge rates** may be linked to **the changes in muscle quality. This was also** reported as a neural strategy by Guo ..."

Comment #7

Lines 87-91: Like comment #5 - please clarify or reference research which shows lower motor unit numbers in older females, and also as suggested here, younger females, compared to males. The references by Guo (2022) and Yacyshyn (2025) do not seem to quantify motor unit numbers. Like above, if it is the number of decomposed units the authors are referring to please specify how that supports the statements made here.

Thank you for the comment. We have removed this sentence as it is indeed ambiguous and does not lend to the message of the perspective.

Comment #8

Paragraph 3 lines 60-91: The authors reference several articles which use HDEMG or intramuscular EMG to investigate neuromuscular function. Despite our limitations of not implementing HDEMG or iEMG techniques, perhaps the authors may like to consider highlighting some of the strengths and the originality of our study and how it supports these previous findings. For example, we included several concomitant measures across all phases of the female lifespan which these studies did not, including sex hormone concentrations, lean skeletal muscle cross-sectional area and intramuscular fat (via pQCT), specific force, and evoked torque and electromyography outcomes. As currently written, the strengths of our study seem lost here and focus is on the omission of more direct motor unit investigation.

Thank you for the comment we have adjusted below to highlight the positive aspects of the paper. We were in no way trying to diminish the paper importance. With word restrictions it was perhaps too ambiguous as we feel this will inspire future work in this area which we referred to as the 'first step'. Please find below the clarification and inclusion of the great number of measures used in the paper.

Pages 3-4, lines 61-75: "...concentrations within each given cohort. **Therefore, the article by O'Bryan and associates (2025) fills a large knowledge gap in this area of study and will hopefully serve as a catalyst for further research in this area.**

The article by O'Bryan and associates (2025) is a significant contribution to fill an important knowledge gap in the literature regarding the adult female lifespan. The large data set across many participants (N=88), representing the stages of the menopause (peri, pre and post), and many measures provide clarity in the neuromuscular function of this understudied group. The study included several associated measures across all phases of the adult female lifespan including sex hormone concentrations, lean skeletal muscle cross-sectional area, intramuscular adipose tissue, specific force, evoked torque and electromyographic outcomes. These measures provide scoping insights into the complexities involved in investigating this vastly understudied population. As identified by the authors, and not their sole measure of neuromuscular function, the use of global electromyography amplitude as a measure..."

Comment #9

Lines 92-95: Please amend this sentence as we did not state that peripheral changes overcome reduced muscle quality. MVC, e1RM, RTD, PT10 and PT100 all decreased during the fourth decade, but changes in specific MVC and intramuscular adipose tissue were not significant (although IMAT tended to increase from 30-70 years). Specific e1RM did decrease with age but it was linear and not obvious to the menopausal transition.

Thank you for the comment, please find below our amended sentence.

Page 5, lines 98-100: "...female adult lifespan, **demonstrating the onset of neuromuscular degeneration from the onset of menopause initially targets intrinsic peripheral muscle function prior to functional declines**. This paper ..."

Comment #10

Line 62 and 95-96: We feel that "first-step" and "starting point" somewhat undersells the depth of our research. We agree that direct investigation into motor unit behavior using HDEMG is necessary (again, which we already stated in our article), however, most other suggested aspects listed in the conclusion by the authors were incorporated into our study and were in addition to previous research including lean skeletal muscle mass, muscle quality (via intramuscular fat infiltration and specific force), muscle function (with voluntary and evoked torque and electromyography measures using twitch interpolation plus one-repetition maximum), sex hormone concentrations, and the menopausal transition. Our study is the first to adopt this extensive and elaborate approach. Perhaps the authors may like to consider amending the language so it is more positive and highlighting more strengths of our study.

Please see the above response to comment #8. Again, we apologize for the misinterpretation of our poor choice of words. Additionally see below for further clarity.

Page 5, lines 100-106: "...This paper **provides a significant contribution and fills several knowledge gaps in the literature** ~~provides a starting point for future research~~ in this vastly understudied area. Future research will need to address ~~the this noted~~ association between hormonal changes during the menopause and the associated changes in muscle loss and muscle quality **and with the addition of** how these changes may influence motor unit behaviour. Additionally, these changes in hormone status and motor unit characteristics **should will need to** be compared ..."

Comment #11

Line 117-119: The title of the final revised and published article is 'The Contribution of Age and Sex Hormones to Female Neuromuscular Function Across the Adult Lifespan'.

Please excuse our mistake. Please find below the correction:

Page 5, lines 128-130: "O'Bryan SJ, Critchlow A, Fuchs C, Hiam D & Lamon S (2025). **The contribution of age and sex hormones to female neuromuscular function across the adult lifespan. ~~The effect of age and sex hormones on female neuromuscular function across the adult lifespan.~~** doi.org/10.1113/JP287496."

Dear Dr Inglis,

Re: JP-P-2025-289013R1 "Unravelling the complexities of neuromuscular function in females throughout the adult lifespan."
by J. Greig Inglis and Helio V Cabral

Thank you for submitting your manuscript to The Journal of Physiology. It has been assessed by a Reviewing Editor and by 1 expert referee. You are now invited to respond to the review comments and submit a revised version for further consideration.

The review comments are copied at the end of this email.

Please address all the points raised and incorporate all requested revisions or explain in your Response to Referees why a change has not been made. We hope you will find the comments helpful and that you will be able to return your revised manuscript within 2 weeks. If you require longer than this, please contact journal staff: jp@physoc.org.

REVISION CHECKLIST:

We look forward to receiving your revised submission.

Yours sincerely,

Richard Carson
Senior Editor
The Journal of Physiology

EDITOR COMMENTS

Reviewing Editor:

We thank the authors for their response to the reviewers comments. After second round of revision, the reviewer raised further comments, specifically on the voluntary and evoked force responses. In the current version of the manuscript, this seems to be not adequately addressed and correct.

REFEREE COMMENTS

Referee #1:

We thank the authors for their respectful and considered response to our first round of comments and changes made to the manuscript. Most of our additional comments are minor, but one statement made in the revised version does not align with our findings and should be amended. Specific comments include:

Comment #1

Line 56-59: If the authors are referring to the rapid decline in force-generating capacity after the 8th decade in humans, then this is well known, however, as written it reads as though the authors are suggesting that dramatic hormonal changes in menopause may result in the rapid decline in force-generating capacity after the 8th decade and that this point is well known, which we don't believe there is evidence to support. Perhaps the authors may like to re-word this section to clarify their point. We also suggest the authors re-consider use of the word "fact" in line 59 as this suggests there is substantial and validated evidence that demonstrates cause and effect relationships.

Comment #2

Line 80: Perhaps adding 'HDEMG' at the end of 'neuromuscular function' here would better guide the reader to seeing the differences between these studies and our own.

Comment #3

Line 86-88: The paper by Guo (2025) did not report associations between muscle quality and impaired neuromuscular function. The authors better convey the findings from that study in lines 93-96, and may like to consider if this sentence is needed. If the authors keep this sentence, then perhaps they could consider re-wording. The year of this reference is also missing.

Comment #4

Line 88-91: We show accelerated reductions in all our voluntary and evoked force responses during the menopausal transition period. This finding is emphasized throughout our article including the key points, however the authors have written here that "they did not find any significant changes in force output" [during menopause] which should be amended.

Comment #5

Line 91-96: Could the authors please specify that although we did suggest higher MUFR may be required to overcome reduced muscle quality, we also stated in our paper the need for future research to directly investigate this association, as neither our paper or the paper by Guo (2025) measured both outcomes and investigated any potential associations.

END OF COMMENTS

EDITOR COMMENTS

Reviewing Editor:

We thank the authors for their response to the reviewers comments. After second round of revision, the reviewer raised further comments, specifically on the voluntary and evoked force responses. In the current version of the manuscript, this seems to be not adequately addressed and correct.

Thank you for your comment, we feel we have fully addressed all the issues below to satisfy the Reviewer's comments.

REFEREE COMMENTS

Referee #1:

We thank the authors for their respectful and considered response to our first round of comments and changes made to the manuscript. Most of our additional comments are minor, but one statement made in the revised version does not align with our findings and should be amended. Specific comments include:

Thank you for your response. Please find below the additional revisions to your comments in full.

Comment #1

Line 56-59: If the authors are referring to the rapid decline in force-generating capacity after the 8th decade in humans, then this is well known, however, as written it reads as though the authors are suggesting that dramatic hormonal changes in menopause may result in the rapid decline in force-generating capacity after the 8th decade and that this point is well known, which we don't believe there is evidence to support. Perhaps the authors may like to re-word this section to clarify their point. We also suggest the authors re-consider use of the word "fact" in line 59 as this suggests there is substantial and validated evidence that demonstrates cause and effect relationships.

We thank the reviewer for your comment and agree that this wording is ambiguous and potentially misleading as indeed they are addressing two separate issues (aging and hormone status). Please find the correction below.

Page 3, lines 56-57: "...a greater loss in force-generating capacity of the muscle in humans. ~~which is even more rapid after the 8th decade in humans.~~ Despite this ~~well-known fact~~, pre-..."

Comment #2

Line 80: Perhaps adding 'HDEMG' at the end of 'neuromuscular function' here would better guide the reader to seeing the differences between these studies and our own.

We thank the reviewer for the comment and have specified that the neuromuscular control in this study was assessed by HDsEMG.

Page 4, lines 79-80: ... women and men, **assessed by intramuscular electromyography and HDsEMG**, that suggest ...”

Comment #3

Line 86-88: The paper by Guo (2025) did not report associations between muscle quality and impaired neuromuscular function. The authors better convey the findings from that study in lines 93-96, and may like to consider if this sentence is needed. If the authors keep this sentence, then perhaps they could consider re-wording. The year of this reference is also missing.

Thank you for the comment. We agree that indeed that the message of the Guo et al paper is better conveyed at the end of the paragraph and have removed this sentence.

Page 4, lines 83-86: “...joint laxity (Inglis & Gabriel, 2021). ~~Guo and colleagues also reported that myosteatosis, the infiltration of fat in the musculature, is greater in older adults, more so in adult females, and is associated with impaired neuromuscular function.~~ As seen...”

Comment #4

Line 88-91: We show accelerated reductions in all our voluntary and evoked force responses during the menopausal transition period. This finding is emphasized throughout our article including the key points, however the authors have written here that "they did not find any significant changes in force output" [during menopause] which should be amended.

Thank you for the comment and we apologize for this mistake. Please find below the amendment.

Page 4, lines 87-90: ...there is a reduction in muscle quality seen in the increase in intramuscular adipose tissue **which was supported by the accelerated reductions in voluntary and evoked force responses during the menopausal transition period** ~~however, they did not find any significant changes in force output.~~ O'Bryan and...”

Comment #5

Line 91-96: Could the authors please specify that although we did suggest higher MUFR may be required to overcome reduced muscle quality, we also stated in our paper the need for future research to directly investigate this association, as neither

our paper or the paper by Guo (2025) measured both outcomes and investigated any potential associations.

Thank you for your comment, we have made the necessary amendment to include your suggesting that future research should directly investigate this association.

Page 4, lines 91-93: "...rates may be linked to the changes in muscle quality and that future research should address this association. Similarly, this was also reported..."

Dear Dr Inglis,

Re: JP-P-2025-289013R2 "Unravelling the complexities of neuromuscular function in females throughout the adult lifespan."
by J. Greig Inglis and Helio V Cabral

We are pleased to tell you that your paper has been accepted for publication in The Journal of Physiology.

Yours sincerely,

Richard Carson
Senior Editor
The Journal of Physiology

If you would like to receive our 'Research Roundup', a monthly newsletter highlighting the cutting-edge research published in The Physiological Society's family of journals (The Journal of Physiology, Experimental Physiology, Physiological Reports, The Journal of Nutritional Physiology, and The Journal of Precision Medicine: Health and Disease), please click this link, fill in your name and email address and select 'Research Roundup':

<https://www.physoc.org/journals-and-media/membernews>

- You can help your research get the attention it deserves! Check out Wiley's free Promotion Guide for best-practice recommendations for promoting your work at: www.wileyauthors.com/eeo/guide. You can learn more about Wiley Editing Services which offers professional video, design, and writing services to create shareable video abstracts, infographics, conference posters, lay summaries, and research news stories for your research at: www.wileyauthors.com/eeo/promotion.

The Corresponding Author will receive an email from Wiley with details on how to register or log-in to Wiley Authors Services where you will be able to place an order

EDITOR COMMENTS

Reviewing Editor:

We thank the authors for incorporating all relevant comments and congratulate them on a well improved manuscript.

REFeree COMMENTS

Referee #1:

We thank the authors for considering our comments and making the appropriate amendments. We have nothing further to add.